# Telomere Length as a New Risk Marker of Early-Onset Colorectal Cancer

**DOI:** 10.3390/ijms24043526

**Published:** 2023-02-09

**Authors:** Abel Martel-Martel, Luis A. Corchete, Marc Martí, Rosario Vidal-Tocino, Elena Hurtado, Edurne Álvaro, Fernando Jiménez, Marta Jiménez-Toscano, Francesc Balaguer, Gonzalo Sanz, Irene López, Sergio Hernández-Villafranca, Araceli Ballestero, Alfredo Vivas, Sirio Melone, Carlos Pastor, Lorena Brandáriz, Manuel A. Gómez-Marcos, Juan J. Cruz-Hernández, José Perea, Rogelio González-Sarmiento

**Affiliations:** 1Institute of Biomedical Research of Salamanca (IBSAL-SACYL), University of Salamanca-CSIC, Paseo de San Vicente, 58-182, 37007 Salamanca, Spain; 2Molecular Medicine Unit, Department of Medicine, University of Salamanca, 37007 Salamanca, Spain; 3Institute of Molecular and Cellular Biology of Cancer (IBMCC), University of Salamanca-CSIC, 37007 Salamanca, Spain; 4Department of Surgery, Vall d’Hebron University Hospital, 08035 Barcelona, Spain; 5Medical Oncology Department, Hospital Universitario de Salamanca, 37007 Salamanca, Spain; 6Department of Surgery, Hospital Universitario Gregorio Marañón, 28007 Madrid, Spain; 7Department of Surgery, Hospital Universitario Infanta Leonor, 28031 Madrid, Spain; 8Department of Surgery, Hospital Galdakao-Usansolo, 48960 Vizcaya, Spain; 9Department of Surgery, Hospital del Mar, 08003 Barcelona, Spain; 10Department of Gastroenterology, Hospital Clínic de Barcelona, Institut d’Investigacions Biomèdiques August Pi i Sunyer (IDIBAPS), Centro de Investigación Biomédica en Red de Enfermedades Hepáticas y Digestivas (CIBEREHD), University of Barcelona, 08036 Barcelona, Spain; 11Department of Surgery, Hospital Clínico San Carlos, 28040 Madrid, Spain; 12Department of Surgery, Hospital MD Anderson, 28033 Madrid, Spain; 13Department of General Surgery and Digestive System, Fundación Jiménez Díaz Hospital, 28040 Madrid, Spain; 14Department of Surgery, Hospital Universitario Ramon y Cajal, 28034 Madrid, Spain; 15Department of Surgery, Hospital Universitario 12 de Octubre, 28041 Madrid, Spain; 16Department of Surgery, Hospital Universitario Fundación Alcorcón, 28922 Madrid, Spain; 17Department of Colorectal Surgery, Clínica Universidad de Navarra, 28027 Madrid, Spain; 18Department of Surgery, Hospital Universitario General de Villalba, 28400 Madrid, Spain

**Keywords:** colorectal cancer, early-onset colorectal cancer, risk, screening, telomere length

## Abstract

Early-onset colorectal cancer (EOCRC; age younger than 50 years) incidence has been steadily increasing in recent decades worldwide. The need for new biomarkers for EOCRC prevention strategies is undeniable. In this study, we aimed to explore whether an aging factor, such as telomere length (TL), could be a useful tool in EOCRC screening. The absolute leukocyte TL from 87 microsatellite stable EOCRC patients and 109 healthy controls (HC) with the same range of age, was quantified by Real Time Quantitative PCR (RT-qPCR). Then, leukocyte whole-exome sequencing (WES) was performed to study the status of the genes involved in TL maintenance (*hTERT*, *TERC*, *DKC1*, *TERF1*, *TERF2*, *TERF2IP*, *TINF2*, *ACD,* and *POT1*) in 70 sporadic EOCRC cases from the original cohort. We observed that TL was significantly shorter in EOCRC patients than in healthy individuals (EOCRC mean: 122 kb vs. HC mean: 296 kb; *p* < 0.001), suggesting that telomeric shortening could be associated with EOCRC susceptibility. In addition, we found a significant association between several SNPs of *hTERT* (rs79662648), *POT1* (rs76436625, rs10263573, rs3815221, rs7794637, rs7784168, rs4383910, and rs7782354), *TERF2* (rs251796 and rs344152214), and *TERF2IP* (rs7205764) genes and the risk of developing EOCRC. We consider that the measurement of germline TL and the status analysis of telomere maintenance related genes polymorphisms at early ages could be non-invasive methods that could facilitate the early identification of individuals at risk of developing EOCRC.

## 1. Introduction

Early-onset colorectal cancer (EOCRC, age < 50 years at diagnosis) incidence has steadily increased over the past few decades. Nowadays, EOCRC accounts for approximately 10% of the total colorectal cancer (CRC) cases diagnosed each year, and it is also showing a significant rise in associated mortality [1]. The pathogenesis of EOCRC is well-characterized among individuals with hereditary CRC. However, 80% of EOCRC cases do not harbor mutations in the genes associated with CRC predisposition [2]. These early onset sporadic tumors are characterized by the development of early metastasis, worse prognosis, and higher aggressiveness. These patients are usually diagnosed at very advanced stages because current screening programs do not consider this age subgroup to be part of the average-risk population. In fact, the European Society of Gastrointestinal Endoscopy (ESGE) only recommends screening for CRC before 50 (beginning at 40 y/o) when defined criteria for familial CRC are fulfilled (at least 2 CRC first-degree relatives (FDR), or at least one FDR with CRC younger than 50 years old) [3]. Thus, the need for developing new markers that can be included in average-risk population CRC prevention strategies to detect individuals at risk for developing EOCRC is undeniable.

Telomerase activity has been postulated as a key factor in cancer development, in terms of cellular immortalization through telomeric elongation as well as in aging and senescence through telomere shortening. In healthy cells, erosion of telomere length (TL) eventually leads to regulated cell senescence and apoptosis. However, in abnormal cells, continued cell division results in telomeric shortening that can lead to end-to-end fusion of chromosomes and chromosomal instability. Therefore, telomere shortening is a process of aging associated with genetic instability [4]. Given this progressive and cell division-dependent erosion, TL is often used as a marker of the replicative history of somatic cells. Although telomeric shortening is usually attributed to defects in DNA replication, it is clearly accelerated by many other factors, such as oxidative stress, replicative stress, and inflammation. From an epidemiological point of view, measurement of TL in cohorts and research studies is of high utility for correlating with demographics, behaviors, health indicators, and other molecular markers. In fact, it has been described that TL measurement of leukocytes can be used as a surrogate marker of relative TL in many other tissues where a correlation with somatic cells exists, since telomerase activity is repressed in most human tissues [5]. Thus, the telomeric shortening observed in germ cells would not be biologically restored in the other tissues at any time; thus, it would correlate with the TL of somatic cells [5].

However, studies on germline cells focusing on CRC telomere dynamics have reported controversial results, with a recent study indicating that both long and short telomeres are associated with an increased risk of CRC [6]. For this reason, we aimed to explore whether TL could be a useful tool in CRC screening programs, particularly in the population under 50 years of age.

## 2. Results

### 2.1. Telomeric Shortening Causes Predisposition to the Development of EOCRC

First, we set out to analyze whether TL variability could be associated with EOCRC. To this end, the absolute TL of leukocytes from 87 microsatellite stable EOCRC patients (excluding those cases with germline pathogenic variants in CRC susceptibility genes) and 109 healthy controls (HC) with the same range of age, obtained from a cohort of healthy patients controlled in Primary Care Units, was quantified by Real Time Quantitative PCR (RT-qPCR). We observed that TL was significantly shorter in the EOCRC group than in the control group (EOCRC mean: 122 kb vs. HC mean: 296 kb; *t*-test, *p* < 0.001) (Figure 1A), which suggests that telomeric shortening could be associated with EOCRC susceptibility. In addition, this association between telomere shortening and the risk of developing EOCRC was maintained across different ages and diagnosis groups: 121 kb vs. 318 kb (<29 years), 101 kb vs. 275 kb (30–39 years) and 118 kb vs. 313 kb (40–50 years); *t*-test, *p* < 0.001 (Figure 1B). Nevertheless, no significant differences (ANOVA test; *p* > 0.05) were found by a TL statistical analysis carried out according to the clinical features of EOCRC patients (Appendix A).

### 2.2. hTERT, POT1, TERF2, and TERF2IP Gene Polymorphisms Are Associated with Telomeric Shortening in EOCRC

We also studied the status of the genes involved in the TL maintenance and the shelterin complex (*hTERT*, *TERC*, *DKC1*, *TERF1*, *TERF2*, *TERF2IP*, *TINF2*, *ACD,* and *POT1*) in leukocytes from 70 sporadic EOCRC cases from the original cohort by whole-exome sequencing (WES). Although we did not find any pathogenic variant that could explain the telomeric shortening observed in our series, we found several SNPs in the *hTERT* (rs79662648), *POT1* (rs76436625, rs10263573, rs3815221, rs7794637, rs7784168, rs4383910, and rs7782354), *TERF2* (rs251796 and rs344152214) and *TERF2IP* (rs7205764) genes with a differential distribution in our cohort compared to the healthy controls, who were enrolled in different databases (Table 1).

## 3. Discussion

Nowadays, it is critically relevant, especially in high-incidence geographic areas, to identify markers that allow us to select the population at risk of developing EOCRC, as most individuals younger than 50 years are not considered for CRC screening strategies, as they are part of the average-risk population [2]. In fact, it has been reported that older age at the time of screening colonoscopy is associated with an increased rate of detection of adenomatous polyps and CRC [7]. This highlights the need for adherence to guidelines to prevent the development of CRC, as well as the identification of useful markers to facilitate the early detection of this disease.

For this reason, we analyzed the absolute leukocyte TL from 87 microsatellite-stable EOCRC patients and 109 healthy individuals in the same age range, showing that germline TL was significantly shorter in EOCRC than in healthy individuals (*p* < 0.001). We also found an association between several SNPs of *hTERT* (rs79662648), *POT1* (rs76436625, rs10263573, rs3815221, rs7794637, rs7784168, rs4383910, and rs7782354), *TERF2* (rs251796 and rs344152214) and *TERF2IP* (rs7205764) genes and the susceptibility to develop this disease. Therefore, our data suggest that telomeric shortening could be associated with a higher risk of developing EOCRC. In this context, it has recently been described that CpGs undergoing aging-related methylation drift were significantly altered in EOCRC, as well as that accelerated aging has been found in normal mucosa from people with EOCRC [8]. Among the cellular factors that correlate with aging, one of the most remarkable is TL. Thus, TL has been established as a useful and easily measurable marker by RT-qPCR [9,10,11,12], although there are other approaches for its quantification, such as terminal restriction fragment (TRF) length measurement by Southern blot [13,14]. In this regard, a study on germline telomere shortening by Lisa Boardman et al. showed that longer telomeres increased the risk of developing EOCRC, whereas shorter telomeres were associated with late-onset CRC [15]. Although these findings contrast with ours, there are methodological differences between both studies: our samples were collected at diagnosis, before any treatment, while in Boardman’s study, samples were collected at any time in the following 2 years. Thus, many patients could have received adjuvant therapy prior to peripheral blood (PB) extraction. In this case, the chemotherapy could have induced the renewal of hematopoietic cells whose TL would be higher. It has also been reported that CRC patients have shorter telomeres in the tumor than in the adjacent mucosa, with no differences between telomere length in PB and the respective tumor samples. An association between telomeric shortening and the presence of metastasis has also been established [16]. Therefore, germline telomere shortening could help to identify not only the population at risk of developing EOCRC, but also individuals who could present a more aggressive phenotype of the disease.

In addition, we studied the status of the genes involved in the TL maintenance and the shelterin complex (*hTERT*, *TERC*, *DKC1*, *TERF1*, *TERF2*, *TERF2IP*, *TINF2*, *ACD,* and *POT1*) in leukocytes from 70 sporadic EOCRC cases from the original cohort using the WES method. We found an association between different SNPs in the *hTERT*, *POT1*, *TERF2,* and *TERF2IP* genes, as well as the susceptibility of developing EOCRC. Interestingly, 7 out of 11 SNPs were found in the *POT1* gene, whose pathogenic mutations have already been related to melanoma, glioma, or CRC susceptibility [17]. However, none of these polymorphisms have previously been linked to cancer risk, except the *POT1* rs7794637 polymorphism, which is associated with breast cancer risk [18]. For that reason, it would be important to consider the status of these *POT1* gene SNPs, as well as their functional impact on telomere maintenance regarding EOCRC risk and/or carcinogenesis. *POT1* forms part of a multiprotein complex that regulates TL and protects chromosome ends from chromosomal instability and abnormal segregation. Consequently, alterations in this protein could lead to a dysfunction of the shelterin complex, which would result in telomere shortening [17]. Although telomere maintenance has also been associated with carcinogenesis, it has been demonstrated that cellular senescence generates a situation of premature aging that also facilitates tumor development and malignant progression [19]. In our cohort, the loss of telomerase activity leading to leukocyte telomeric shortening and, therefore, to premature cellular aging could cause a predisposition to the development of EOCRC.

## 4. Materials and Methods

### 4.1. Patient Selection

For this study, we selected 87 EOCRC cases without any germline pathogenic variant in CRC hereditary genes. They were selected from the prospective multicenter study defined as the Spanish Early-Onset Colorectal Cancer Consortium (SECOC) [20]. All patients were diagnosed before the age of 50 years, without history of inflammatory bowel disease and with a histopathological diagnosis of adenocarcinoma. Clinicopathological features were determined from a detailed review of the medical records. Variables included gender, age at CRC diagnosis, body mass index (BMI) at diagnosis, tumor stage at diagnosis, tumor location, histological features (grade of differentiation, mucinous component, “signet ring” cells), multiple primary neoplasms (synchronous and metachronous CRC) and familial cancer history (Appendix A). As a control group, we selected 109 healthy individuals aged under 50 years from the Salamanca Primary Care Unit, with no history of neoplastic disease nor pathologies associated with TL disorders. This study was authorized and registered in January 2021 by the Human Research Ethics Committee of the Hospital Universitario Fundación Jiménez Díaz (PIC012-21_FJD; PI José Perea) and was approved by each center enrolled in the SECOC cohort as follows: MD Anderson Cancer Center Madrid, Hospital Universitario Ramón y Cajal, Hospital Clínico San Carlos, Hospital General Universitario Gregorio Marañón, Hospital Universitario 12 de Octubre, Hospital Universitario Fundación Jiménez Díaz, Hospital Universitario Infanta Leonor, Hospital Universitario Fundación Alcorcón, Hospital Universitario General de Villalba, Hospital Universitario de Salamanca, Hospital Clinic de Barcelona, Hospital del Mar, Hospital Universitario de Bellvitge, Hospital Universitario Vall d’Hebron, Hospital Universitario Galdakao-Usansolo, Hospital Universitario de León, and Clínica Universidad de Navarra. We confirm that all participants have provided their written informed consent in this study. All methods were carried out in accordance with the relevant guidelines and regulations.

### 4.2. DNA Isolation and Telomere Length Analysis by Real Time Quantitative PCR (RT-qPCR)

DNA was obtained from PB leukocytes using the phenol–chloroform method. PB sample collection from EOCRC patients was performed at diagnosis, prior to the initiation of any type of treatment. All DNA samples were stored in Eppendorf tubes at 20 °C to prevent their progressive degradation and potential contamination. The TL of the leukocytes was measured by RT-qPCR using the Absolute Human Telomere Length Quantification qPCR Assay Kit (ScienCell, Catalog #8918, Carlsbad, CA, USA), following the manufacturer’s instructions. This technique allows the initial amount of DNA coding for telomerase (TEL) and a single copy reference gene (used as an endogenous control) to be quantified simultaneously. The difference in the amount of DNA quantified represents the relative TL of each sample. To analyze these relative changes, a reference fragment of known TL (provided by the manufacturer) was added to each assay, allowing the absolute quantification of the TL of each sample. Triplicate reactions were carried to minimize variability. The TEL and SCR fragments were amplified using 10 ng in 2 µL of DNA, 1 µL of each specific primer, and 10 µL of the FastStart SYBR Green Master Mix. The amplification program was as follows: 10 min at 95 °C followed by 40 cycles at 95 °C for 15 s, 52 °C for 30 s, and 60 °C for 1 min. Finally, the Ct (2^−∆∆Ct^) comparative method was used to calculate the relative DNA amount of each amplicon. This assay was performed in a 96-well plate and the detection was carried out in the Step-One Plus Real-Time PCR system (Applied Biosystems, Waltham, MA, USA).

### 4.3. Library Preparation and Sequencing

Whole-exome sequencing was performed in FACS-sorted clonal PCs and paired T lymphocytes from 70 patients with EOCRC. Exonic sequences were enriched using the Ion Ampliseq Exome kit (Life Technologies, Carlsbad, CA, USA), sequenced with a 151 × 2 bp read length and 150× depth coverage using semiconductor technology (Ion Proton, Carlsbad, CA, USA), and were then analyzed with the Ion Reporter and Torrent Suite Software (Life Technologies). Prior to analysis, FASTQ files were assessed for read quality using the FASTQC tool (v.11.9) [21]. Raw reads were mapped to the human genome GRCh37 (hg19) version from the UCSC using the BWA-MEM alignment algorithm (v.0.7.17) [22]. The resulting SAM files were subsequently processed to improve the efficiency of the successive analysis. First, PCR duplicates were marked, and SAM files were transformed to a coordinate-sorted BAM file using the MarkDuplicatesSpark tool from the GATK analysis toolkit (v.4.1.4.0) [23]. Second, read groups were added to these BAM files using the AddOrReplaceReadGroups tool from Picard (v.2.9.0-1). For the next step, we performed a base quality score recalibration on our BAM files to eliminate errors made by the sequencer when the accuracy of each base was estimated. This last step was performed by combining the BaseRecalibrator and ApplyBQSR GATK tools. Once our BAM files were recalibrated, we carried out a set of analyses to retrieve the mutational status from our samples. Somatic single nucleotide variants (SNV) and INDELs were called at using the Mutect2 GATK tool, through its matched mode. Cross-sample contamination was estimated using the CalculateContamination tool, and, finally, SNVs were filtered through the FilterMutectCalls option. All SNVs and INDELs were annotated by the Funcotator GATK tool and were considered to be positive when called by ≥10% variant reads. Variant pathogenicity was estimated through the Varank (v.1.4.3) tool [24]. SNV frequency and distribution in our cohort were compared to two public healthy subject databases, NHLBI-ESP (Exome Variant Server, NHLBI GO Exome Sequencing Project (ESP), Seattle, WA, USA; available at https://evs.gs.washington.edu/EVS (accessed on 10 June 2022) [25] and GNOMAD; available at http://gnomad.broadinstitute.org (accessed on 10 June 2022) [26], through the ANNOVAR tool [27], which was accessed in June 2022.

### 4.4. Statistical Analyses

Categorical variables were expressed as number of cases and percentage, and these were compared using Pearson’s Chi Square (χ^2^) test. Comparing continuous with categorical variables, either Student’s *t*-test or ANOVA was used for independent samples and the Mann–Whitney U test was used for continuous variables (all of which were expressed as mean values plus/minus standard deviation (SD)). SPSS version 23.0 (IBM) was used for statistical analyses. A *p*-value less than 0.05 was established as the threshold to consider the differences as statistically significant.

## 5. Conclusions

In conclusion, our results suggest that telomeric shortening could be associated with EOCRC susceptibility. Moreover, the analysis of the *POT1* gene showed that different SNPs of this gene are more represented in the EOCRC series. For that reason, we consider that the measurement of germline TL and *POT1* gene polymorphisms at early ages could be non-invasive methods that could facilitate the early identification of individuals at risk of developing EOCRC. This would also facilitate the identification of those individuals who do not meet the currently established CRC screening criteria, allowing us to anticipate the starting age of screening for average-risk CRC. Thus, this could justify including TL and POT1 SNPs as a new markers in CRC prevention strategies, after consequent validation in other young population cohorts.

## Figures and Tables

**Figure 1 ijms-24-03526-f001:**
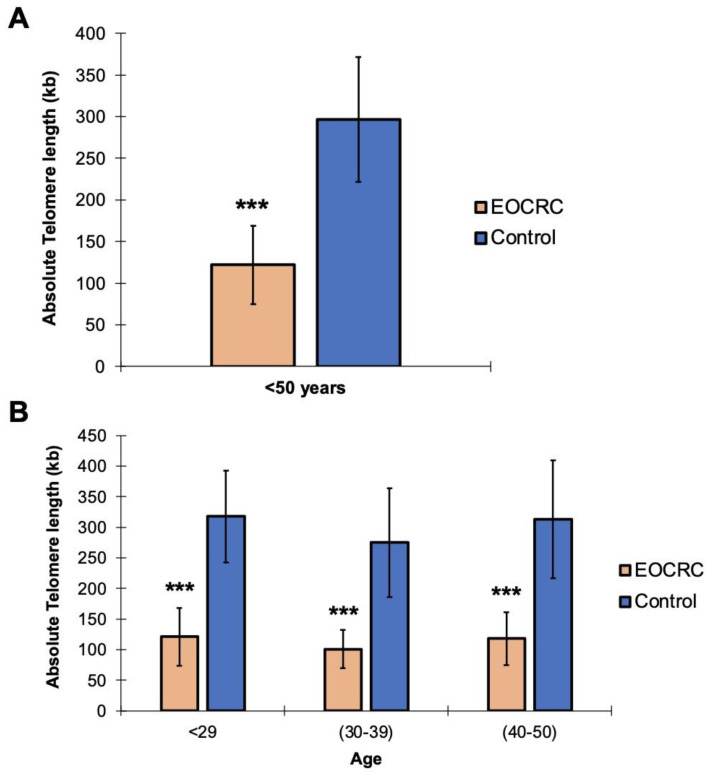
Absolute telomeric length quantification by RT-qPCR in DNA from EOCRC patients’ peripheral blood and that of healthy individuals. (**A**) Representation of absolute germline TL in EOCRC patients and healthy individuals below 50 years of age. (**B**) Representation of germline absolute TL in EOCRC patients and healthy individuals by age groups: <29 years, 30–40 years, and 40–50 years. Absolute telomeric lengths are represented in kilobases (kb). Data are shown as the means ± SD, where *** indicates *p* < 0.001 for EOCRC vs. Control group. The TL of all patients was measured in triplicate.

**Table 1 ijms-24-03526-t001:** Telomere maintenance-related gene SNPs showing statistical differences between patients with sporadic EOCRC and two healthy subjects’ databases. SNP: single nucleotide polymorphism; REF: reference allele; ALT: alternative allele; Freq.: frequency of ALT allele; NHLBI GO-ESP: National Heart Lung and Blood Institute Grand Opportunity Exome Sequencing Project; EA: European ancestry; GNOMAD NFE: Genome Aggregation Database of non-Finish European; FDR: false discovery rate; N/D: not determined; NA: not applicable. Significant differences are marked in bold.

				EOCRC (*n* = 70)	NHLBI GO-ESP (EA-6500 Samples)		GNOMAD AF_NFE (56885 Samples)	
Gene	SNP	REF	ALT	Freq.	Freq.	FDR	Freq.	FDR
*hTERT*	rs79662648	C	G	0.086	0.036	0.0568	0.036	**0.0486**
*POT1*	rs76436625	T	C	0.257	N/D	NA	0.108	**0.0008**
*POT1*	rs10263573	A	T	0.571	0.405	**0.0112**	0.396	**0.0050**
*POT1*	rs3815221	G	A	0.571	0.405	**0.0112**	0.396	**0.0050**
*POT1*	rs7794637	T	C	0.900	0.697	**0.0006**	0.690	**0.0002**
*POT1*	rs7784168	T	C	0.529	0.305	**0.0006**	0.316	**0.0007**
*POT1*	rs4383910	A	C	0.829	N/D	NA	0.548	**0.0000**
*POT1*	rs7782354	C	T	0.829	N/D	NA	0.573	**0.0001**
*TERF2*	rs251796	A	G	0.514	0.298	**0.0006**	0.304	**0.0007**
*TERF2*	rs34415214	G	A	0.157	0.062	**0.0106**	0.061	**0.0050**
*TERF2IP*	rs7205764	T	C	0.643	N/D	NA	0.506	**0.0323**

## Data Availability

The datasets used and/or analyzed during the current study are available from the corresponding author upon reasonable request.

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
