# Peer review of "Telomere Length as a New Risk Marker of Early-Onset Colorectal Cancer"

_ijms, 2023, doi:10.3390/ijms24043526_

Round 1
Reviewer 1 Report
In this paper Martel-Martel A et al aimed at exploring whether an aging factor such telomere length could be a useful tool in Early-onset colorectal cancer screening, on a total of 87 EOCRC cases compared with 109 healthy patients. Since telomere length was significantly shorter in EOCRC than in healthy individuals (EOCRC mean: 54 122 kb vs HC mean: 296 kb; p<0.001), they concluded that the measurement of germline telomere length could be non-invasive methods for early diagnosis of EOCRC.
It is for sure an arguing issue, however some points should be addressed in a revision.
- In the introduction section it is indicated that telomerase activity has been postulated as a key factor in cancer development. Authors should clear explain why it is preferable the measurement of telomere length to the more simple dosage of telomerase activity (10.1016/j.arr.2013.12.006 – reviewer is not an Author of suggested reference)
- Paragraph 2.2. Why you choose to measure the telomere length of leukocytes by RT-qPCR? Why you did not measure telomere lengths using Southern blot analysis of the mean terminal restriction fragments (TRF) lengths, as described by Kimura et al? 10.1038/nprot.2010.124
- Results. Please state if you run a statistical analysis (regression) attempting to identify other confounding factors that could affect your results (i.e. sex, stage of disease, etc…). If not, I strongly suggest run that analysis and to properly report the results.
- I suggest you run analysis aiming at identify the TL according to different “age at diagnosis” groups
- Please start the discussion section with the main findings of your study.
- Discussion section appears very poor as regard the discussion of your data with the current literature. Additionally, it should be indicated the clinical useful of your results, explaining the importance of TL as diagnostic tool.
- it would be of immense interest the longitudinal study on TL changes at different levels of disease (diagnosis, surgery, adjuvant treatments, etc…). have you these data?
Author Response
In this paper Martel-Martel A et al aimed at exploring whether an aging factor such telomere length could be a useful tool in Early-onset colorectal cancer screening, on a total of 87 EOCRC cases compared with 109 healthy patients. Since telomere length was significantly shorter in EOCRC than in healthy individuals (EOCRC mean: 122 kb vs HC mean: 296 kb; p<0.001), they concluded that the measurement of germline telomere length could be non-invasive methods for early diagnosis of EOCRC. It is for sure an arguing issue, however some points should be addressed in a revision.
Point 1: In the introduction section it is indicated that telomerase activity has been postulated as a key factor in cancer development. Authors should clear explain why it is preferable the measurement of telomere length to the more simple dosage of telomerase activity (10.1016/j.arr.2013.12.006 – reviewer is not an Author of suggested reference).
Response 1: We appreciate this suggestion from the Reviewer. This point has been clarified in paragraph 1.2 of the introduction section.
Point 2: Paragraph 2.2. Why you choose to measure the telomere length of leukocytes by RT-qPCR? Why you did not measure telomere lengths using Southern blot analysis of the mean terminal restriction fragments (TRF) lengths, as described by Kimura et al? 10.1038/nprot.2010.124
Response 2: We thank the Reviewer for raising this point. In this study, the RT-qPCR method was used for telomere length measurement due to the high throughput of the technique, the facility to access to the material needed to carry it out, as well as the possibility to measure up to 15 patients every 2 hours. In addition, this technique only required 10 ng of DNA per sample for each assay. This allowed us to use a considerable amount of DNA to perform other assays, such as WES. In contrast, the Southern Blot technique would require up to 3 µg of DNA/sample as well as the use of radioactive compounds for telomere measurement. Finally, the RT-qPCR telomere measurement kit used in this study (detailed in the methods section) is targeted against a specific sequence, whereas the Southern Blot could detect and measure both canonical and non-canonical telomere components.
Point 3: Results. Please state if you run a statistical analysis (regression) attempting to identify other confounding factors that could affect your results (i.e. sex, stage of disease, etc…). If not, I strongly suggest run that analysis and to properly report the results.
Response 3: We thank the Reviewer for this comment. A statistical analysis has been performed according to the clinical data reported in Supplementary Table 1. The results obtained have been described in the results section of the Main Text as well as in the Supplementary Table 1.
Point 4: I suggest you run analysis aiming at identify the TL according to different “age at diagnosis” groups.
Response 4: We appreciate this suggestion from the Reviewer. A statistical analysis has been carried out according to the age at diagnosis of the disease. The results have been included in the Main Text results section as well in the Figure 1B.
Point 5: Please start the discussion section with the main findings of your study.
Response 5: We thank the Reviewer for this comment. We have included the appropriate modifications at the beginning of the discussion to detail the main results of our study. Because this manuscript has a communication format, we have preferred to discuss based on two main sections to facilitate the comprehension of the results: telomeric shortening in EOCRC and the association of telomere maintenance related genes and the susceptibility to develop EOCRC.
Point 6: Discussion section appears very poor as regard the discussion of your data with the current literature. Additionally, it should be indicated the clinical useful of your results, explaining the importance of TL as diagnostic tool.
Response 6: We thank the Reviewer for raising this point. The discussion content and length have been adapted to the communication format proposed by the editor. Likewise, the pertinent changes have been introduced in the discussion section of the Main Text with the aim of detailing our results and highlighting the importance of telomere measurement as a useful tool in screening programs, as well as it is pointed out in the Conclusions sections. Our data support the possible usefulness of this marker in CRC screening strategies, which would help not to universalize colonoscopy as well as to identify possible subjects at risk of developing CRC that are not covered by current strategies.
Point 7: It would be of immense interest the longitudinal study on TL changes at different levels of disease (diagnosis, surgery, adjuvant treatments, etc…). have you these data?
Response 7: We appreciate this suggestion from the Reviewer. In this work, we have analyzed the telomere length of EOCRC patients before initiating any type of treatment, since the main objective of this study is to find a risk marker for the development of the disease that will facilitate its screening and diagnosis. However, we take this suggestion with special interest for future studies.

Reviewer 2 Report
In this manuscript, Martel et al. present a study on telomere length as a new risk marker of early-onset colorectal cancer. The authors assessed the absolute leukocytes telomere length from 87 microsatellite stable EOCRC patients and 109 healthy controls with the same range of age by RT-qPCR. Then, leukocyte whole-exome sequencing was performed to study the status of the genes involved in the telomere length maintenance (hTERT, TERC, DKC1, TERF1, TERF2, TERF2IP, TINF2, ACD and POT1) in 70 sporadic EOCRC cases from the original cohort, which showed that telomere length was significantly shorter in EOCRC than in healthy individuals (EOCRC mean: 122 kb vs HC mean: 296 kb; p<0.001), suggesting that telomeric shortening could be associated with EOCRC susceptibility. They found a significant association between several SNPs of hTERT (rs79662648), POT1 (rs76436625, rs10263573, rs3815221, rs7794637, rs7784168, rs4383910, rs7782354), TERF2 (rs251796, rs344152214) and TERF2IP (rs7205764) genes and the risk of developing EOCRC, indicating that the measurement of germline telomere length and the status analysis of telomere maintenance related genes polymorphisms at early ages could be non-invasive methods that could facilitate the early identification of individuals in risk of developing EOCRC. This manuscript is interesting, and the experiments are well thought out and expertly executed, I think it can be accepted for publication.
Author Response
Point 1: In this manuscript, Martel et al. present a study on telomere length as a new risk marker of early-onset colorectal cancer. The authors assessed the absolute leukocytes telomere length from 87 microsatellite stable EOCRC patients and 109 healthy controls with the same range of age by RT-qPCR. Then, leukocyte whole-exome sequencing was performed to study the status of the genes involved in the telomere length maintenance (hTERT, TERC, DKC1, TERF1, TERF2, TERF2IP, TINF2, ACD and POT1) in 70 sporadic EOCRC cases from the original cohort, which showed that telomere length was significantly shorter in EOCRC than in healthy individuals (EOCRC mean: 122 kb vs HC mean: 296 kb; p<0.001), suggesting that telomeric shortening could be associated with EOCRC susceptibility. They found a significant association between several SNPs of hTERT (rs79662648), POT1 (rs76436625, rs10263573, rs3815221, rs7794637, rs7784168, rs4383910, rs7782354), TERF2 (rs251796, rs344152214) and TERF2IP (rs7205764) genes and the risk of developing EOCRC, indicating that the measurement of germline telomere length and the status analysis of telomere maintenance related genes polymorphisms at early ages could be non-invasive methods that could facilitate the early identification of individuals in risk of developing EOCRC. This manuscript is interesting, and the experiments are well thought out and expertly executed, I think it can be accepted for publication.
Response 1: We thank the Reviewer for this comment.

Round 2
Reviewer 1 Report
- Authors provided the proposed suggestions.
- Anyway, the point n. 2:
Point 2: Paragraph 2.2. Why you choose to measure the telomere length of leukocytes by RT-qPCR? Why you did not measure telomere lengths using Southern blot analysis of the mean terminal restriction fragments (TRF) lengths, as described by Kimura et al? 10.1038/nprot.2010.124
Response 2: We thank the Reviewer for raising this point. In this study, the RT-qPCR method was used for telomere length measurement due to the high throughput of the technique, the facility to access to the material needed to carry it out, as well as the possibility to measure up to 15 patients every 2 hours. In addition, this technique only required 10 ng of DNA per sample for each assay. This allowed us to use a considerable amount of DNA to perform other assays, such as WES. In contrast, the Southern Blot technique would require up to 3 µg of DNA/sample as well as the use of radioactive compounds for telomere measurement. Finally, the RT-qPCR telomere measurement kit used in this study (detailed in the methods section) is targeted against a specific sequence, whereas the Southern Blot could detect and measure both canonical and non-canonical telomere components.
- I understand the response, but I suggest to add this method as "limitation" in the discussion section (10.1111/acel.12289 the reviewer is not the author of suggested paper).
Author Response
Authors provided the proposed suggestions. Anyway, the point n. 2:
Point 2: Paragraph 2.2. Why you choose to measure the telomere length of leukocytes by RT-qPCR? Why you did not measure telomere lengths using Southern blot analysis of the mean terminal restriction fragments (TRF) lengths, as described by Kimura et al? 10.1038/nprot.2010.124
Response 2: We thank the Reviewer for raising this point. In this study, the RT-qPCR method was used for telomere length measurement due to the high throughput of the technique, the facility to access to the material needed to carry it out, as well as the possibility to measure up to 15 patients every 2 hours. In addition, this technique only required 10 ng of DNA per sample for each assay. This allowed us to use a considerable amount of DNA to perform other assays, such as WES. In contrast, the Southern Blot technique would require up to 3 µg of DNA/sample as well as the use of radioactive compounds for telomere measurement. Finally, the RT-qPCR telomere measurement kit used in this study (detailed in the methods section) is targeted against a specific sequence, whereas the Southern Blot could detect and measure both canonical and non-canonical telomere components.
Point 1 (round 2): I understand the response, but I suggest to add this method as "limitation" in the discussion section (10.1111/acel.12289 the reviewer is not the author of suggested paper).
Response 1 (round 2): We thank the Reviewer for this comment. This point has been clarified in the second paragraph of the discussion section (lines 165-167), which describes the usefulness of RT-qPCR for telomere length measurement in other studies, as well as the possibility of using Southern Blot alternatively.
